# Flexible Phased Antenna Arrays: A Review

**DOI:** 10.3390/s25154690

**Published:** 2025-07-29

**Authors:** Waleef Ullah Usmani, Francesco Paolo Chietera, Luciano Mescia

**Affiliations:** Department of Electrical and Information Engineering, Politecnico di Bari, Via E. Orabona 4, 70125 Bari, Italy; francescopaolo.chietera@poliba.it (F.P.C.); luciano.mescia@poliba.it (L.M.)

**Keywords:** flexible antennas, phased antenna arrays, additively manufactured antennas

## Abstract

Flexibility in phased antenna arrays open the world of new applications. Such arrays can conform to different shapes while ensuring performance in harsh conditions. The purpose of this study is to perform a detailed comparative analysis of numerous studies of flexible phased antenna arrays. This work summarizes the main manufacturing techniques and the commonly used materials with their properties. It also outlines the key challenges and future trends in the development of flexible phased antenna arrays. The paper concludes with research recommendations to address the identified technical issues.

## 1. Introduction

During the past few decades, Phased Arrays (PAs) have been prioritized over traditional arrays due to specific features such as fast and reliable beamforming, the ability to generate multiple beams with minimum interference, the ability to rapidly steer beams electronically, coverage of large areas without mechanically moving parts, and the ability to dynamically adapt to different operational needs [1].

### 1.1. Historical Background

PAs faced challenges of cost, size, and component limitations in the implementation of Electronically Scanned Antenna Arrays (ESAAs) before 1980s [2,3,4]. These systems implemented Passive Electronically Scanned Arrays (PESAs) instead of Active Electronically Scanned Arrays (AESAs). The implementation of AESAs was made possible after the inclusion of Gallium Arsenide (GaAs)-based circuits at a monolothic scale during 1980s. Monolithic phase shifters utilizing GaAs technology have been actively incorporated in applications such as electronic warfare and Satellite Communications (SATCOM) ever since [5,6]. The advancement to incorporate multiple functionalities on a single chip was utilized for phase and gain control across different frequency bands [7,8,9], consequently reducing the size of the arrays. But the lower yield and higher costs of the solution posed a hurdle to realize complex architectural choices for array design. The limitations were addressed through the implementation of silicon-based Radio-Frequency Integrated Circuits (RFICs), which offer enhanced integration, reduced size, and improved performance in PA, leading to an increase in the number of their applications. Silicon-based RFIC allowed the realization of fully integrated multi-element PA systems for the first time. The idea of a fully integrated PA system is to have a Transmitter (TX) and Receiver (RX) capable of beamforming, gain, and phase control, with integrated on-chip power amplifiers, and Low-Noise Amplifiers (LNAs) [10,11,12,13].

### 1.2. Silicon-Based RFIC Technology and Diverse Applications

The advancement of silicon-based RFIC technology led to fully integrated PA systems operating across multiple frequency bands since the late 2000s. Notable examples include Ka-band TX/RX on-chip systems [14], Q-band PA TX based on RFIC [15], and system-level implementations in X/Ku-band using single silicon chips [16], demonstrating complex PA architectures for various applications [17]. This evolution enabled compact and power-efficient solutions for 5G devices [18], where machine learning techniques have also been explored for PA synthesis [19]. PA have also been incorporated in Wireless Power Transfer (WPT) systems, such as near-field neuro-modulation [20] and channel-based intelligent WPT with phase control for beam steering [21]. Further applications include medical imaging and meteorological remote sensing [22,23]. Beam steering and amplitude tapering previously required element-wise calibration, but this was addressed in 2017 by a fully functional RFIC-based PA with orthogonal phase and amplitude control for 5G communications [24]. These developments were later adopted in SATCOM applications [25,26]. Silicon-based RFICs enabled efficient and compact PA systems with wide-band and wide-scan performance, scalability, high Effective Isotropic Radiated Power (EIRP), and improved Cross-Polarization Discrimination (XPD) [27]. The use of RFIC beamformer chips with Wilkinson combiner networks on Printed Circuit Board (PCB) allowed polarization-agile PA systems with high polarization purity and simultaneous multi-beam control [28,29,30]. Integration of multiple functions on a single chip reduced discrete components, improving form factor and lowering cost [31]. In 2014, a wafer-scale approach combining all components on the same silicon wafer further enhanced array compactness [32]. By addresing size, cost, efficiency, and scalability, silicon-based RFIC technology has enabled the transition from bulky, mechanically steered systems to compact AESAs, opening opportunities in consumer electronics, automotive, and wearable technologies.

### 1.3. Flexibility in Phased Arrays

After the utilization of fully integrated silicon-based RFIC technology in PA, the next step was to introduce flexibility in PA systems. Flexible Phased Arrays (FPAs) are significantly important in terms of applications like avionics, automotive, warfare, aerospace or even wearable devices. Practical utilization of FPAs started in early 2000s with the development of active PA antenna using membrane technology [33] and Micro-Electro-Mechanical System (MEMS) micro-strip phase shifters on a flexible Liquid Crystal Polymer (LCP) substrate [34]. They led to the development of fully integrated PA system on a single chip with RF-MEMS switches on a flexible LCP substrate for the first time in 2008 [35]. When flexibility is introduced, the resulting deformation changes the geometry of the antenna array. For the preservation of the radiation pattern, a phase compensation mechanism was developed for a self-adapting FPA [36]. In a similar work, more recently, another self-adapting array was developed which was able to dynamically compensate for performance degradation due to real-time deformations [37]. Manufacturing of FPAs has evolved over time with the advancement in Additive Manufacturing (AM) technology. For example, Inkjet Printing (IP) was used for the fabrication of an FPA system using a flexible Kapton substrate in 2013 [38]. AM precisely deposits materials on flexible substrates, paving way for lightweight, and adaptable systems. Despite the revolutionary potential, the evolution of technology still faces some significant challenges such as material limitations, mechanical deformation effects, thermal management, and design complexity.

### 1.4. Problem Statement

Although the FPA concept encompasses a wide spectrum of shape-adaptive array architectures, much of the existing literature has focused on individual implementations or specific technologies, rather than providing a comprehensive comparison across different solutions. Given the rapid evolution of enabling materials, manufacturing processes (both subtractive and additive), and design methodologies, a comparative analysis is needed to identify the key differentiators and limitations of each approach. Such analysis can be useful to guide future developments by clarifying performance metrics, integration challenges, and areas where further innovation is required. Accordingly, the following Section 2 presents different manufacturing techniques exploited for producing FPA; Section 3 highlights the materials used for their realization; Section 4 focuses on the major challenges identified in the field of FPA; while Section 5 presents a comparative analysis of the various technologies based on key performance parameters. Finally, Section 6 concludes the review, trying to guide the reader through the future trends of this promising field.

## 2. Manufacturing Techniques

Advancements in FPA systems are largely driven by the development of different manufacturing techniques with the aim of achieving lightweight, flexible, and high performance in the designs. In this section, a structured analysis of the different technologies used in the manufacturing of FPA is presented.

### 2.1. Additive Manufacturing

One of the most widely utilized AM techniques in PAs is IP. IP is a contactless deposition method where drops of functional ink are deposited onto a substrate, enabling the creation of conductive traces and antenna elements directly onto flexible substrates. The process is composed of several steps including ink formulation, substrate preparation, drop-on-demand printing, and post-processing. Conductive inks, usually based on Silver Nanoparticles (SNPs), are utilized with flexible substrates such as PET and polyimide (Kapton). To achieve the conductive films, nanoparticles must be interconnected through as sintering process (post-processing). Ink design aims to minimize sintering temperature for compatibility with flexible substrates. IP is a low-cost fabrication process which enables rapid design and customization while minimizing the material waste, and it is compatible with bendable substrates, but post-processing may impose limitations on substrate choices. IP is extensively utilized to fabricate conductive traces, antenna elements, and interconnections. This technology can also be combined with other AM methods, such as Fused Filament Fabrication (FFF), which is widely used for prototyping conformal or geometrically complex, and potentially flexible, antenna structures [39,40]. For example, a wideband circularly polarized (CP) patch antenna array was fabricated on Polyethylene terephthalate (PET) film using SNP-based inks for the first time [41]. Similarly, a two-dimensional PA employed IP carbon nanotube thin-film transistor (CNT-TFT)-based phase shifters on Kapton substrates [38]. IP was combined with FFF and Stereolithography (SLA) to design a wearable FPA with integrated microfluidic cooling channel for effective thermal management [42]. In Figure 1a, the overall system layout is shown, while Figure 1b illustrates the difference in uniformity of the conductive inkjet-printed ink between the sample exposed to Ultra-Violet Ozone (UVO) treatment and the one not exposed. This work showed robust performance under bending and thermal stress.

In another solution for SATCOM and 5G millimeter wave applications, a tile-based solution of FPA was presented, exploiting IP for the fabrication of both the individual tiles and a flexible tiling layer [43]. Utilization of IP is not limited to only SATCOM and 5G millimeter wave applications, but its usefulness is also evident in case of Radio Frequency Identification (RFID) and sensing applications [44]. IP together with other AM techniques is useful for realizing low-profile and lightweight antenna elements. For example, a low-profile single-substrate multi-metal layer Ultra Wide Band (UWB) antenna was developed using Additive Manufacturing Electronics (AME) which is a particular kind of IP technology able to deal with both conductive SNP ink and dielectric (acrylate) materials, printing them in a multilayer configurations using an automated system [45]. The multilayer structure to achieve UWB is shown in Figure 2. The figure shows a three-layer structure of patch antenna. It consists of a driven patch and two stacked patches with truncated corners. The aim for the truncated corners is to induce CP.

This was the first demonstration of UWB, low-profile antennas using AME and integrating conductive and dielectric layers. In another work, fully IP PA was fabricated on a Kapton flexible substrate. The array was integrated with CNT-TFTs as switching elements for beam steering [46]. This was one of the earliest demonstrations of lightweight and conformal PA, but IP was already being utilized in fabrication of ultra-thin flexible substrates [47]. Another implementation of AM-based UWB antenna for FPAs was presented in [48]. A stacked patch antenna configuration was used to achieve wide bandwidth while the top and bottom patches were printed with IP onto a PET substrate. UWB was also achieved in the development of an ultrathin and flexible array using a three-layer laminate design and Frequency Selective Surface (FSS) [49]. IP has been widely utilized in numerous works due to its low cost and scalability. In addition to IP, additive fabrication facilitates the creation of complex, multifunctional PAs. Additive fabrication of multilayer flexible PCBs using Pyralux© All Polymide (AP) laminates was utilized in the fabrication of a 256-element FPA system with the aim of mechanical flexibility [50]. The front and back view of the final prototype of FPA system with 256-elements is presented in Figure 3. A similar design was presented for a lightweight FPA system integrating advanced electromagnetic components with self-deployable and self-contained power system. The PA used Pyralux®, AP8545R flexible polyimide substrate for mechanical flexibility [51]. In another work, an AM tile-based conformal array was designed by utilizing a computer vision based calibration of the array which was experimentally validated under symmetric and asymmetric bending configurations [52].

### 2.2. Hybrid Rigid–Flexible Designs

The combination of rigid components with flexible substrates offers a practical balance between structural stability and mechanical adaptability. This concept is known as rigid–flex antenna array (RFAA), which is a hybrid architecture. It employs rigid radiators fabricated on substrates like Rogers®RO4350, integrated with flexible polyimide circuitry. These designs facilitate rollable and deployable configurations, such as for small satellite (SmallSat) applications. Rigid areas provide mechanical support for antenna elements and soldered active components, while flexible circuits route RF signals. In some circumstances, capacitive feed mechanisms eliminate the need for conductive vias, simplifying assembly while maintaining performance during repeated mechanical deformation [53]. In a similar attempt, a combination of flexible DuPont Pyralux®AP PCBs (flexible substrate) and rigid Rogers Duroid®5880 PCBs was recently utilized in a novel checkerboard configuration. The flexible material and rigid one were arranged in an alternating pattern to enable flexibility in both the *x* and *y* axes without compromising overall mechanical integrity. Novel checkerboard design provided the system with the ability to conform to a cylindrical geometry with a radius of 50 mm, expanding the range of potential applications [54]. Another work focusing on hybrid design for SATCOM demonstrated a deployable PA system where a 32-element Ka-band active PA was implemented on a flexible four-layer LCP substrate [55]. This trend of utilizing deployable PAs continued with more novelties, and in 2023, a heterosegmented LCP substrate was used combining six-layer and two-layer LCP substrates to achieve deployability and flexibility [56]. Other novel strategies have also been used to counter different challenges in FPAs. For example, to achieve a scalable production, a novel strategy of thermal drawing process was utilized for the design of a photonics based FPA for the first time. It integrated photonics and RF systems embedded in Polyetherimide (PEI) fiber carriers. This novel design achieved broadband matching and stable performance with flexibility across the ultra-high-frequency (UHF) band [57].

The diverse range of manufacturing techniques for FPAs reflected the adaptability of these methods to a large variety of application requirements, but along with the manufacturing techniques, the choice of materials used in the design of FPAs is also critically important.

## 3. Mostly Used Materials

An analysis of various studies on material characterization for FPAs reveals consistent trends in the selection of materials for both conductive and dielectric layers. Copper evidently emerges as the most widely used conductive material, particularly in traditional PCB-based implementations, where its high electrical conductivity and excellent RF performance make it the preferred choice. Its effectiveness also extends to rigid–flex approaches, ensuring low-loss transmission and reliable integration. Although chromium is also reported as a conductive material in some cases, its use is significantly less common [50,51,54,58,59,60,61]. In contrast, SNP ink is gaining attention for its compatibility with AM techniques, especially in the advanced manufacturing of flexible electronics. Despite its lower conductivity compared to copper, SNP remains a viable alternative for RF applications due to its excellent processability and suitability for printed electronics [43,44,62]. Its compatibility with IP processes makes it particularly attractive for complex geometries and conformal surfaces [41,42,63].

In terms of dielectric materials, the analysis highlights two main categories: semi-rigid dielectric substrates and flexible polymers compatible with flexible applications. Among semi-rigid substrates, Rogers® holds a leading position due to a series of patents covering PTFE-based laminates composed of resin mixtures with various fillers, yet free of internal glass fabric. This composition allows controlled mechanical flexibility under specific bending conditions, making them suitable for semi-conformal implementations. In particular, Rogers® RT5880 is extensively used in high-frequency applications due to its low dielectric constant and minimal loss tangent, which are critical for preserving signal integrity and reducing RF losses [50,51,54,59,61]. Comparable performance is also offered by other PTFE-based laminates (with or without woven glass reinforcement) such as Rogers® 4003C, Rogers® 3003, and Taconic® FastFilm™, which combine suitable dielectric characteristics with moderate flexibility [44,62,64]. However, fully flexible substrates are required for highly flexible or deployable applications such as space-deployable arrays. Several polymers have been widely adopted in the reviewed studies. Kapton® HN, a polyimide, combines good dielectric properties with excellent mechanical flexibility and thermal stability. This allows it to conform to non-planar surfaces and endure high-temperature processes without significant degradation [62]. LCP is another commonly used substrate due to its low dielectric constant (≈2.9–3.2) and low loss tangent (≈0.002–0.005), together with its natural flexibility and high dimensional stability under bending [65,66,67]. An example of such a single antenna element fabricated using the LCP flexible substrate and the antenna array fabricated using the LCP substrate demonstrating the bending capability of such a design is presented in Figure 4. Polydimethylsiloxane (PDMS) and PEI fibers have also been explored for their conformability and thermal resistance, making them suitable for integration on cylindrical or deformable surfaces [57,68].

An example of a flexible radiating element and an array configuration is shown in Figure 4. Other notable flexible solutions include DuPont Pyralux®-based laminates, which combine thin polyimide films with copper cladding, enabling multilayer flexible designs with good electrical and mechanical performance. The comparison of these materials, including their mechanical, electrical, and application-driven motivations, is provided in Table 1.

## 4. Major Challenges

Although FPAs offer significant potential for a wide range of advanced applications, their benefits do not come without critical challenges that must be addressed to ensure the performance of the final device. This section reviews the main issues identified across relevant studies.

### 4.1. Mechanical Deformation and Its Impact on Performance

FPAs are often deployed on either curved surfaces or dynamically deforming surfaces, consequently, introducing phase distortions, impedance mismatches, and degradation of beamforming accuracy. Arrays mounted on deformable surfaces suffer from phase variation as a result of geometric distortions. The array factor describes how the geometrical combination of multiple antenna elements affects the overall radiation pattern. It represents the impact of the geometry of the array, amplitude and phase of the excitation signals applied to each radiating element. Generally, in an FPA, the radiating elements are properly arranged on a curved surface. So, the array factor is given by(1)$$AF\left(\theta ,\phi \right)=\sum_{n=1}^{N}a_{n}e^{jk\left[x_{n}\left(u-u_{s}\right)+y_{n}\left(v-v_{s}\right)+z_{n}\left(w-w_{s}\right)\right]}$$
with(2)$$\begin{array}{cc} \; & u=\sin\theta \cos\phi \\ \; & u_{s}=\sin\theta_{s}\cos\phi_{s}\\ \; & v=\sin\theta \sin\phi \\ \; & v_{s}=\sin\theta_{s}\sin\phi_{s}\\ \; & w=\cos\theta \\ \; & w_{s}=\cos\theta_{s} \end{array}$$

In (Equation 1)–(Equation 2), $a_{n}$ is the amplitude of the *n*th radiating element, k=2π/λ is the wave number at the frequency *f*, $x_{n}$, $y_{n}$, and $z_{n}$ are the position coordinates of the *n*th element relative to the center oh the global coordinate system, θ and ϕ are the elevation and azimuth angles pertaining an arbitrary observation point, $\theta_{s}$ and $\phi_{s}$ are the scanning angles, $u_{s}$, $v_{s}$, and $w_{s}$ are the scanning directions in the *u*-*v*-*w* space.

Unlike the planar array, in the FPA, the *z*-axis coordinate of each radiating element changes according to its position over a 3D surface. As a result, the radiation pattern of FPA differs from that of the planar one even with the same number, shape, and spacing of the radiating elements. Phase of the signal is determined by its position relative to a reference point, so the variation due to the mechanical deformation affects the array factor. This variation can alter the conditions for constructive and destructive interference leading to the distortion in main lobe and side lobe level (SLL). This issue can be dealt with compensation mechanisms and calibration techniques for the impact of mechanical deformation on electromagnetic performance. For example, two studies utilized mutual coupling measurements to sense shape distortions and compensate for phase and one of them implements a self-calibration mechanism [69,70]. Similarly, another study addressed this challenge by integrating flexible resistive sensor and analog circuitry to monitor surface deformations and adjust phase shifters for real-time radiation pattern correction in dynamically changing conditions [36]. The problem of shape reconstruction was dealt with, in a different way with the introduction of Frequency Shift Key (FSK) radar approach to reconstruct the geometry of FPAs by estimating inter-element spacing [62]. For the preservation of the radiation pattern, one approach was implemented using phase in 2022 [71]. Moreover, frequent bending and rolling of FPAs can degrade electrical and mechanical integrity. Repetitive mechanical stress can induce small cracks in printed metallic traces leading to the disruption of the current flow, reducing the signal integrity which affects the gain and beam steering capability. Substrates such as PET or Rogers laminates experience fatigue over repeated flexing/bending, which can result in loss of elasticity. A solution to this challenge can be the use of a rigid–flex architecture which was briefly discussed in the previous sections. It combines rigid substrate tiles mounted on flexible dielectric laminates, so defining the precise region in which the device can bend and obtaining flexible and lightweight antenna arrays. For example, an RFAA maintained performance after 50 roll–unroll cycles in the application in small satellite lightweight apertures [53]. A similar study showed an array conformed to a cylindrical geometry, maintaining performance under deformation by alternating rigid–flex configuration [54]. With the aim of completely eliminating the rigid phase shifter circuitry to improve flexibility and durability under bending, a novel technique was implemented which utilized silver-coated ferrite particles to achieve tunable phase shifts by aligning the particles using a static magnetic field. The use of static magnetic fields to activate phase shifters eliminated the need for external biasing, significantly reducing complexity [59]. In another approach, a bending radius as small as 5 mm was achieved in 2021 by utilizing Taconic®fastFilm™ to develop a low-loss flexible PCB-based TX/RX module that maintains high RF performance under bending [64]. In general, RFAA is an efficient solution for FPA. Beamformer IC can be attached to the rigid parts of the device without the problem of bending degradation. While this is an efficient solution, maintaining electrical continuity between the flexible and rigid sections without degrading performance is still a key challenge. Furthermore, flexibility in RFAA can only be achieved as a global property with tiles that locally remain rigid but can conform globally due to the presence of the rigid tiles. Moreover, bending among the two axes (around a sphere) still remains challenging. The discussed design strategies as well as pros and cons of their phase compensation methods are summarized in Table 2. Future trends of research to face the mechanical deformation challenges lie in the development of advanced algorithms for phase compensation based on Field Programmable Gate Array (FPGA) technology.

### 4.2. Electrical Performance Limitations

Achieving consistent electrical performance, including low insertion loss, wide bandwidth, and stable impedance matching, is a core challenge in FPA designs. Flexible substrates have different dielectric properties compared to traditional rigid materials, which lead to variations in signal propagation speed, impedance, and power loss. Flexible materials such as polyimide and Kapton® exhibit higher dielectric losses compared to rigid substrates. This was evident in the design of PAs printed by IP which encountered a significantly performance-degrading insertion loss of 8.17 dB [38]. The utilization of CNT-TFTs as switching elements for phase shifters introduced a high insertion loss reducing overall system performance. Similarly, in the design of self-adapting FPA, gain degradation (loss of 0.6–1.8 dB) due to phase shifter insertion loss was found [36]. The impact of insertion loss was minimized in different studies by optimizing the design parameters. For example, curved microstrip lines exhibited only a 0.3 dB increase in insertion loss compared to planar configurations, indicating negligible performance degradation [64]. Moreover, a highly efficient PA system with insertion loss below 0.1 dB was demonstrated through direct on-package fabrication, eliminating traditional connectors and thus shortening the signal path [42].

Bandwidth is also an important electrical performance parameter. Some applications require wide-band operation which is often constrained by coupling effects and impedance mismatches, especially on flexible platforms. In an attempt to enhance the bandwidth, stacked-patch design and inverted-F feed structures were utilized to ensure impedance matching and bandwidth enhancement. This solution delivered the widest bandwidth (17.5%) among active bendable PA antennas while maintaining scanning up to 60° [72]. Among other methodologies to achieve a wider bandwidth, an important technique is the use of proper feeding mechanism. In the design of a flexible Ka-band active PA on LCP substrate to support SATCOM applications, a novel feeding mechanism of proximity-fed patch antennas leveraged the proximity effect to achieve a wider bandwidth [67]. With the integration of LCP layers, bandwidth limitations emerge as a major challenge in such applications that is mitigated by using proximity feed design [55]. An interesting field of research regards the use of AME platforms to achieve precise multilayered structures while maintaining full custom design.

Impedance matching in FPA is another critical parameter. In [61], authors designed periodic rectangular microstrip patches with an inset feed. The design reduced mutual coupling between elements by modifying the patch dimensions and utilized a two-level multiple-section feeding network for improved impedance matching and spectral performance.

Wide angle scanning requires the achievement of a stable beam steering over wide angles, but this often results in grating lobes or gain reduction. This hurdle was mitigated by using a simplified phase-shifting network [73]. The suppression of gratin lobes is also possible with another technique of utilization of a triangular lattice configuration, particularly the offset rows in the triangular configuration mitigated the grating lobes, but this resulted in the increased design complexity for modular integration [72].

### 4.3. Fabrication and Manufacturing Challenges

Fabricating FPAs involves significant challenges related to substrate handling, multilayer integration, and precision assembly, all of which are critical for achieving flexibility. In particular, multilayer PCB integration introduces issues such as alignment accuracy and interlayer crosstalk. For example, a deployable phased-array transmitter using a four-layer LCP substrate integrated RF, ground plane, Serial Peripheral Interface (SPI) control, and patch antennas to achieve compactness. However, this configuration revealed the inherent difficulties in maintaining uniformity during the multilayer lamination process [65]. These critical challenges include layer-to-layer alignment sensitivity due to LCP dimensional instability under thermal cycling. This can be mitigated by utilizing low-temperature lamination techniques to reduce the expansion/contraction of LCP or by using pre-conditioned LCP sheets to stabilize thermal expansion prior to assembly.

Scalability was identified as another challenge, and fabrication methods such as IP, while cost-effective, are limited in scalability for large arrays. This can be dealt with by the division of multilayer architecture into separate modules connected through flexible interposers to reduce complexity. Another way is to utilize the modular rigid–flex architectures to isolate sensitive components from deformation zones.

Furthermore, the design of wideband stacked patch antenna array using IP emphasized the challenges in achieving uniformity and consistency in multilayer designs [41]. The fabrication process involved IP on PET. It posed the challenge of Silver conductivity limitation, photonic curing, stack alignment (the ground and radiating layers were printed on separate PET films, necessitating accurate alignment during assembly). IP is an efficient fabrication technique, but it also introduces alignment and attachment challenges [63]. A possible solution for future works can be the utilization of conformal pick-and-place systems with 3D vision feedback to map surface topology and guide placement.

In addition to aligning and attaching ICs on 3D-printed surfaces, addressing the surface roughness of 3D-printed materials to improve the metallization quality posed another challenge [42]. The surface roughness of printed materials (that depends on the printing technology as well as on the working frequency of the final device) has a significant impact on metallization quality. It can cause non-uniform copper deposition, leading to higher surface resistance and degradation of RF performance. Sanding and surface finishing techniques are utilized to counter this to smooth the substrate before electroplating.

### 4.4. Thermal Management

The inclusion of active components, such as RFICs and amplifiers, introduces significant thermal challenges such as heat dissipation in FPAs. Power amplifiers in particular convert a significant amount of power into heat, leading to localized hot spots. Furthermore, thin and flexible substrates (LCP and PET film) used in FPA have limited thermal conductivity, making heat management critical, impeding heat dissipation compared to rigid PCBs or metal-backed substrates. Such temperature variations can cause mechanical stress on the components due to mismatched coefficients of thermal expansion between components. This phenomenon can lead to the risks of delamination or cracking, especially in laminated or 3D-printed structures. Moreover, electrical performance of RFIC can be significantly affected, which can lead to shift in frequency responses, increase in noise figure, and reduction in linearity. Consequently, without efficient thermal management, long-term exposure to high temperatures reduces component reliability. This issue was addressed with various methodologies. For example, microfluidic cooling channels were embedded to reduce the temperature of the IC in short time interval by using water flow [42,63]. Thermal management is also critical in high-power configuration of FPAs. For example, in a 256-element FPA system for WPT, with high-power configuration delivering power wirelessly at a 1 m distance, thermal management was achieved through aluminum heat spreaders in high-power system and flat ceramic spreaders in the flexible system [50]. Similarly, for an FPA system with 16 radiators, Power Synthesis and Control Unit (PSCU) was particularly used to ensure stable voltage and current distribution during operation with sufficient temperature control [51]. Thermal insensitivity is a desirable feature. In another work, a thermally and mechanically insensitive antenna array was developed using a PDMS substrate that overcomes the challenge of thermal interference due to the low value of the thermal coefficient of PDMS substrate by reducing the effects of anisotropic thermal expansion [68]. Different thermally resilient materials and integrated cooling channels integrated with dynamic cooling systems can be viable solutions for efficient thermal management.

### 4.5. Environmental and Operational Challenges

The real-world deployment of FPAs requires not only high electromagnetic performance but also robust operation under various environmental and mechanical stress conditions, including extreme temperatures, humidity, moisture exposure, and mechanical deformation from stretching and bending. So, a crucial area of research focuses on maintenance of consistent performance in face of such challenges. This has led to the exploration of environmentally resilient materials and mechanically adaptive design strategies. As an example, low-loss substrates have been utilized to reduce the dielectric losses under variation of environmental conditions such as moisture absorption and thermal fluctuations [61]. Real-time phase calibration techniques have been introduced to tackle the problem of environmental variability affecting array performance, specifically in beamforming and signal alignment. Such systems have continuous element-wise monitoring and phase readjustment in response to the changes in environmental conditions, leading to dynamic beam correction [50]. In recent years, FPA have been widely utilized in disaster response and field surveillance equipment. In such applications, self-sufficient power solutions are of high significance. One such innovative approach integrated Photovoltaic (PV) concentrators directly with power amplifiers. This enabled a standalone operation. This is considered as a significant advancement for long-duration deployments without access to conventional power infrastructure dedicated to remote deployment [51]. Although not for remote environments, another study was conducted for a smart and wearable textile-based antenna array. The solution incorporated textile-based microstrip antennas with direction of arrival estimation for dynamic beam correction, achieving a low-cost and scalable structure [74].

## 5. Performance Comparison

Following the analysis of the main challenges in FPA design, a comparative study was carried out based on key performance parameters. The results are summarized in Table 3, which provides a compact overview of the surveyed works on FPAs. The table includes multiple parameters essential for evaluating FPA performance: operating frequency (freq), array size, bandwidth (BW), gain (in dB), maximum beam steering angle, insertion loss, beam width, flexibility characteristics, bending radius or curvature (flex curve), bending performance, polarization type (pol), and thermal performance (thermal perf). A multi-dimensional evaluation is necessary due to the variability introduced by different application scenarios. Among the most critical parameters, bandwidth reflects frequency agility, while the maximum steering angle quantifies beam reconfigurability. Insertion loss is analyzed to assess signal power loss and system efficiency. Beam width is relevant for resolution and directionality. Flexibility-related information is divided into how flexibility is achieved (e.g., flexible substrates, additive manufacturing, rigid–flex integration), the achieved bending radius, and the impact of bending on performance. Lastly, the table reports the polarization (e.g., linear or circular) and thermal behavior, including maximum operating temperature, cooling strategies, and any observed performance degradation under thermal stress.

It is possible to extract some information regarding the performance, design choice, and practical implications of FPA based on the comparative analysis across multiple works. FPAs are being designed over a range of frequencies highlighting the diversity in applications such as UHF communications, SATCOM, 5G/6G communications, and radar systems. The size of the arrays also varies from small and compact 1×4 arrays to large 16×16 configurations depending on the type of application. This reflects the trade-off between size of the array, gain of the array, and beam control. Moreover, the gain of the array varies from small values of 1 dB or 6 dB for smaller arrays to higher values as high as 60.5 dB (EIRP) for higher frequencies and large scale designs. Moreover, beam scanning is a major factor in PA, with a substantial number of works with a steering angle range up to ±70°. This indicates strong reconfigurability. The wide scanning range comes with an increase in the value of insertion loss and a degradation in the gain, especially at extreme angles. Based on the type of applications, bandwidth also varies, with some designs supporting UWB operation while others are optimized for narrowband applications. This suggests a trade-off between bandwidth and parameters such as structural complexity and fabrication methodologies. Furthermore, the major focus of this comparative analysis was flexibility. Flexibility in FPA is primarily achieved with the utilization of flexible substrates such as polymide and LCP, sometimes using AM like IP to define the conductive traces on the flexible substrates. Another approach for obtaining flexibility was introduced in the form of hybrid rigid–flex design. Performance under bending is also analyzed for the analysis with bending radii ranging from 3 cm to 50 cm, with most studies having stability in performance under mechanical deformation. Moreover, in some circumstances, phase compensation techniques are implemented to mitigate performance degradation. Ultimately, thermal performance is discussed, particularly for high-power and space applications. Several arrays demonstrate good thermal stability up to 400 °C or are labeled as compatible for space applications, highlighting robustness under extreme conditions. The key trade-offs are between the flexibility and performance parameters such as bandwidth, gain, and beam steering. In cases of high flexibility, gain or bandwidth can be compromised to a certain extent. The field of FPAs has evolved to include high-performance, thermally stable, and reconfigurable antenna arrays with considerable mechanical flexibility. Emerging trends highlight the adoption of AM, microfluidic cooling systems for thermal management, advanced phase compensation techniques, and hybrid rigid–flex designs. The comparative analysis emphasizes the need to balance flexibility, electromagnetic performance, and environmental resilience to meet the requirement for real-world applications.

## 6. Conclusions and Future Trends

This paper analyzed several critical aspects related to the design and development of FPAs. An overview was provided of the most commonly used materials and technologies, as well as the major limitations currently encountered in their implementation.

A first key direction explored involves the use of flexible substrates and AM techniques, such as IP, to realize conductive elements. While this approach is widespread, it typically enables only passive array configurations due to the inherent difficulty in implementing multilayer structures (integrating both conductive and dielectric layers). These multilayers are essential for embedding the control and RF electronics required in active array systems.

Currently, the AME platform Dragonfly by Nanodimensions is the only one that supports 3D-printed multilayer architectures of both conductive and dielectric materials. However, they are limited by the narrow range of compatible dielectric materials and the lack of robust evidence supporting their integration with standard soldering processes which remain fundamental for high-performance RF systems.

From a manufacturing point of view, a promising alternative lies in the use of rigid–flex PCB technologies. These offer a trade-off between structural flexibility and the reliability of conventional fabrication processes. This area is expected to witness significant developments in the near future, especially as the demand for conformal and adaptive antennas increases.

Another key consideration is the impact of mechanical deformation on electromagnetic performance. The non-planar positioning of radiating elements in flexible arrays can degrade radiation characteristics. Addressing this issue likely requires the development of compensation techniques, such as phase correction and real-time electronic reconfiguration, to maintain optimal performance under deformation.

Furthermore, the design of arrays based on unconventional geometries, such as concentric or concentric–hexagonal lattices, opens new opportunities for performance improvement, particularly in terms of SLL mitigation. However, these configurations pose new challenges for the design and implementation of complex feeding networks, which requires dedicated research efforts.

In conclusion, the data presented in this review can serve as a starting point for identifying the target specifications for various application scenarios involving FPAs. They also provide a useful reference for guiding future research activities in this promising technology field.

## Figures and Tables

**Figure 1 sensors-25-04690-f001:**
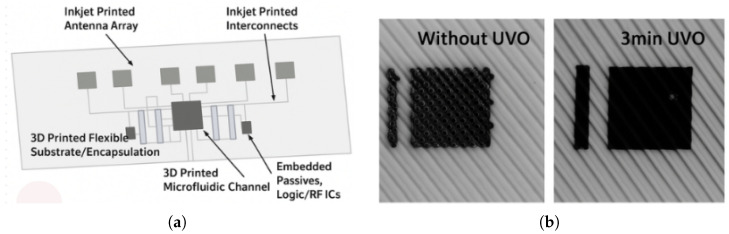
(**a**) Proof-of-concept demonstration of flexible and wearable IP antenna array showing antennas, interconnects, flexible substrate, and microfluidic cooling channel. (**b**) Comparison of IP SNP ink on printed substrates with and without UVO surface treatment. Surface wetting can be enhanced by employing UVO treatment to increase the hydrophilicity of plastic surfaces [42].

**Figure 2 sensors-25-04690-f002:**
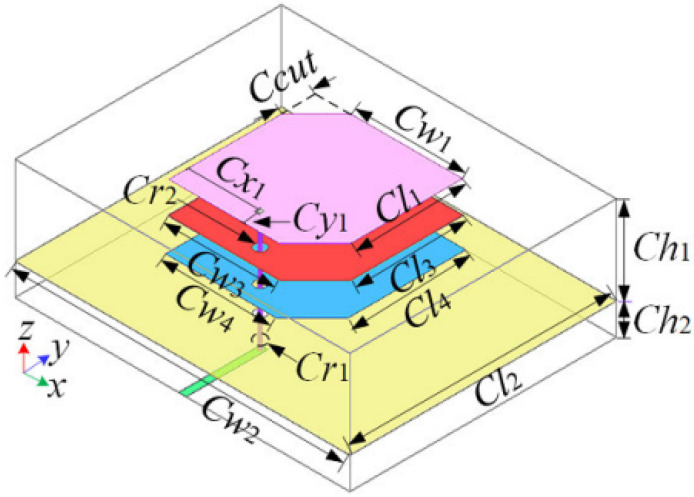
Three-layered structure of patch antenna (one driven patch and two stacked patches with truncated corners) used to ahieve UWB. Technique of truncated corners is used to induce CP [45].

**Figure 3 sensors-25-04690-f003:**
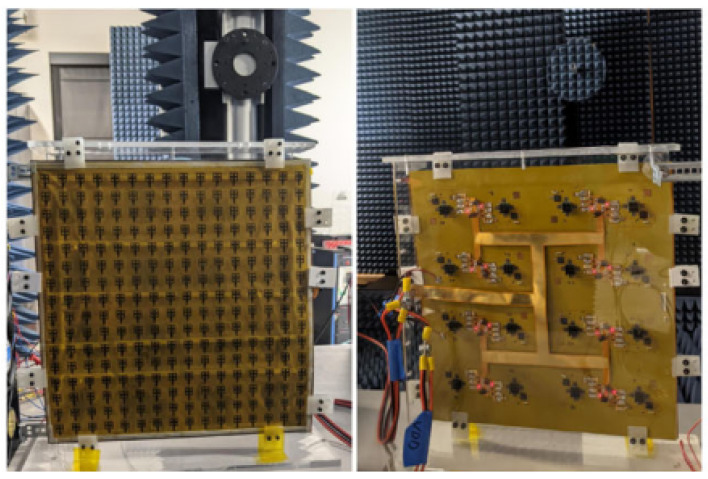
Photograph of the FPA antenna (front and back view). The FPA is equiped with deformation correction and is able to transfer RF- direct current (DC) wireless power transfer of about 80 mW at 1 m away from the transmitter [50].

**Figure 4 sensors-25-04690-f004:**
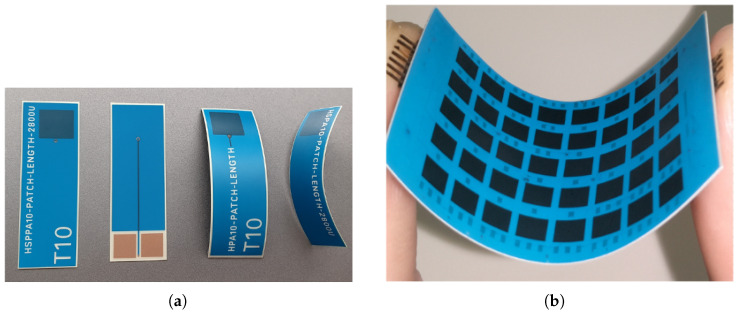
(**a**) Photograph of a single antenna element using flexible LCP substrate with bending demonstration of the single element. (**b**) Bending demonstration of a square lattice planar antenna array made of LCP substrate [65].

**Table 1 sensors-25-04690-t001:** Material properties.

Materials Used	Mechanical Properties	Electromagnetic Properties	Reason for Use
Chromium	Durable, Thin	Highly Conductive	Conductivity
Rogers® RT5880	Semi-flexible, Lightweight	Dielectric constant = 2.2, Loss tangent = 0.0009	Low loss, flexibility
DuPont Pyralux®	Flexible, Lightweight	Dielectric constant = 3.4, Loss tangent = 0.003	Flexible, thin, low profile
Copper	Durable	Conductive, Low resistivity	Conductivity, reliability, compatible with PCB standard manufacturing
LCP	Foldable, Lightweight, Deployable	Dielectric constant = 3.0–3.3, Loss tangent = 0.0049	Low loss, flexible
Rogers® 4003C	Rigid, Lightweight	Dielectric constant = 3.38–3.55, Loss tangent = 0.0027	Low loss
Rogers® 3003	Semi-Flexible, Durable, Lightweight, Thermally-conductive	Dielectric Constant = 3.00, Loss tangent = 0.0013	Low dielectric loss and flexible
SNP ink	Flexible, Printable	Conductive, low resistivity	Printability, thermal stability
Taconic® FastFilm™	Durable, Lightweight	Dielectric constant = 3.55, Loss tangent = 0.0027	Low loss, flexibility
Rogers® 3003	Semi-Flexible, Durable, Lightweight, Thermally conductive	Dielectric Constant = 3.00, Loss tangent = 0.0013	Low dielectric loss and flexibility
Kapton® HN Polyimide	Lightweight, Flexible, Durable, Thermostable	Dielectric constant = 3.4, Loss tangent = 0.002–0.005	Thermal stability
SNP ink	Flexible, Printable	Conductive, low resistivity	Printability, thermal stability
Taconic® FastFilm™	Durable, Lightweight	Dielectric constant = 3.55, Loss tangent = 0.0027	Low loss, flexibility
PP	Lightweight, Robust	Dielectric constant = 2.2–2.34, Loss tangent = 0.0002–0.001	Low loss, robustness
PET	Flexible for small thickness, Durable	Dielectric constant = 3.0–3.4, Loss tangent = 0.02	Cost-effectiveness
PDMS	Flexible, Conformable	Dielectric constant = 2.65, Loss tangent = 0.031	Flexibility and thermal stability
PEI fiber	Flexible, Durable	Dielectric constant = 3.0, Loss tangent = 0.0012	Thermal stability

**Table 2 sensors-25-04690-t002:** Comparison among different designs and phase compensation strategies.

Paper	Design	Phase Compensation	Advantages	Disadvantages
[69]	Folded dipoles on copper ground	Spiral Match algorithm, Semi-Definite Relaxation	No additional sensors, Modular	1D bending, Inaccuracy in non-uniform bends
[70]	Flexible substrate with copper PCB	Mutual coupling used to infer relative phase center (differential phase measurement)	No need for external sensors, Operates under bending, High mechanical flexibility	Variations in antenna/feed network not accounted for
[36]	Microstrip patch array printed on a flexible substrate	Phase compensation based on the data of sensor deformation detection	Low complexity, Pattern recovery under deformation	Limited to predefined geometrical shapes
[62]	Probe-fed patch antennas mounted on an AM acrylonitrile butadiene styrene (ABS) frame	A phase-based distance measurement, FSK-based measurement using phase differences at two frequencies	Sensor-free approach, Applicable to arbitrary deformations	Reduced accuracy at close spacing
[71]	Patch array built on thin flexible composite laminate made of E-glass fiber and a highly conductive mesh-style fabric	Phase shifters are implemented as extended microstrip line lengths, introducing phase delays	High mechanical flexibility, Low ohmic and dielectric losses	Only suitable for fixed deformations
[53]	Rigid–flex flexible membrane for RF and DC routing, with rigid tiles as substrates for antenna and components	No phase compensation but element-level electronic beam steering	Ultra-lightweight, Highly flexible, Active component integration	No active beam steering demonstrated
[54]	Rigid–flex checkerboard structure combining rigid PCBs and flexible substrate	No dynamic phase compensation	Flexible in 2D, Dual polarization	Lacks real-time phase control
[59]	Patch array with integrated magnetic particles based Composite Right/Left-Handed (CRLH) metamaterial phase shifters	Phase control is achieved by tuning the permeability (per-element phase tuning)	Real-time tunability, Conformal flexibility	Complex fabrication, Magnetic field source required
[64]	Double-layer flexible PCB	Integrated digital phase shifters	Low loss under bending, Integrated active electronics	No dynamic phase calibration, Assembly complexity

**Table 3 sensors-25-04690-t003:** Performance comparison.

NO.	Freq. (GHz)	Size	BW (GHz)	Gain (dB)	Steering	Insertion Loss (dB)	Beam Width	Flexibility	Flex Curve	Bending	Pol.	Thermal Perf.
[33]	1.26	2 × 4	0.08	Rx 17, Tx 30	$0^{\circ }$–$15^{\circ }$	–	–	Flexible sub.	–	Good	–	–
[34]	14	–	4	–	–	0.59 (1–bit)	–	Flexible Sub. LCP	–	Curling addressed	–	LCP Properties
[35]	14	2 × 2	≈0.7	7.75	$\pm 12^{\circ }$	4.57	–	Flexible Sub	–	Minimal impact	–	Max 150 °C
[36]	2.45	1 × 4	≈0.02	5.9 (flat)	≈$30^{\circ }$–$45^{\circ }$	2–3	Broad, Narrow	Flexible Sub	$30^{\circ }$–$45^{\circ }$ wedge, 10 cm cylinder	Autonomous correction	–	–
[37]	2.25	1 × 8	100 MHz	14.79	$\pm 40^{\circ }$	–	≈$77^{\circ }$	Flex Sub	13.2 c m	–	CP	–
[38]	5	4 × 4	–	14.6	$\pm 34^{\circ }$	8.17	–	Flexible Sub AM	–	Demonstrated	–	–
[43]	19	2 × 2 tile	–	16	$\pm 50^{\circ }$	–	$15^{\circ }$–$25^{\circ }$	Flexible Sub AM	–	Retain perf.	–	Stable $\approx 400^{\circ }$C
[61]	10	1 × 4	1.3	12.4	$\pm 30^{\circ }$	–	Varies	Semi Flex	20 cm	Phase compensation	–	–
[41]	7.7–8.3	4 × 4	0.6	16.9	–	Varies	–	Flex Sub AM	≥1.2 cm	Gain reduction	RHCP	–
[42]	24–40	–	Wide Band	Max ≈10.09	$\pm 37^{\circ }$	–	Broad	Flex Sub AM	3 cm	Minimal shift	Circular	Microfluidic Cooling
[63]	24–40	–	Wide Band	Max ≈10.09	$\pm 37^{\circ }$	–	Broad	Flex Sub AM	3 cm	Stable for 10,000 bending cycles	Circular	Microfluidic Cooling
[44]	30	5 × 1 patches	0.6	12.5	$\pm 70^{\circ }$	1.17	–	Flex Sub AM	3 cm	Minimal RCS variation	LP (Dual)	Robust
[45]	Sub-6	2 × 2 array	0.47	7.7	–	–	–	Flex Sub 3D	–	–	CP	–
[46]	4.99	1 × 4	–	–	$\pm 27^{\circ }$, $\pm 45^{\circ }$	–	–	Flex Sub	6.5 c m	24 cm	–	–
[47]	4.79–5.04	1 × 4	0.25	4.5	$\pm 20^{\circ }$	–	$5^{\circ }$, HPBW	Flex Sub AM	5 cm, 25 cm	Stable	CP	Moderate Heating
[48]	22.3–42.5	2 × 4	20.2 G Hz	13 dBi	–	–	–	Flex Sub	–	Effective	–	Temp. stable
[49]	2–18	6 × 6	15 GHz	–	–	–	–	Flex Sub	20 cm	–	–	–
[50]	10	30 × 30 cm^2^ sheet, 256 elements	UWB	≈60.5 EIRP	$\pm 10^{\circ }$ azimuth, $\pm 3^{\circ }$ elevation	–	2D Scan	Flex Sub	23 cm	Effective	LP	Temp. stable
[51]	9.4–10.4	4 × 4 tile	1.0	≈37.1 EIRP	$\pm 56.4^{\circ }$ without GLs	–	$14^{\circ }$–$22.5^{\circ }$	Flex Sub	3 cm	Stable	LP (Dual)	Space Compatible
[52]	19.5	2 × 1 and 4 × 1	–	≥7	$-45^{\circ }$ to $+45^{\circ }$	–	–	Flex Sub	up to $45^{\circ }$	–	–	–
[53]	9.4–10.4	16 × 16	0.8	15.2	$\pm 30^{\circ }$	–	Consistent	Rigid Flex	3.8 c m	50 roll–unroll cycles	LP	Space Compatible
[54]	10.7–12.75	8 × 4 (checkerboard)	2.05	16.3	$\pm 60^{\circ }$	–	Consistent	Rigid Flex	5 cm	Measured on cylinder	LP (Dual)	–
[55]	28	32 elements	≈0.8	EIRP ≈41	$\pm 50^{\circ }$	–	$+10^{\circ }$ at edges	Flex Sub	Deployable	–	LP	–
[56]	26.25	8 × 8	≈0.5	46.7	$\pm 10^{\circ }$ to $+20^{\circ }$	–	$7^{\circ }$	Flex Sub	Foldable	$\pm 10^{\circ }$ bent	–	–
[57]	0.42–0.45	Linear Array	Broad UHF	≈5	$\pm 15^{\circ }$	<3	$19.5^{\circ }$	Flex Sub	–	–	Dual polarized	Minimal Deformation
[58]	24	10 × 10	0.67	29.4	$149.8^{\circ }$ (θ), $120^{\circ }$ (ϕ)	–	Broadside, $\pm 30^{\circ }$	Flex Sub AM	4.5 c m	Stable	LP	Consistent
[59]	2.45	1 × 4	0.5	9.52	$\pm 30^{\circ }$	≈1	Broadside, $\pm 30^{\circ }$	MP CRLH Phase Shifter	30 cm, 10 cm	Minimum variation	LP	Minimum Deformation
[60]	5.8	1 × 4	5–6	7.38 ($0^{\circ }$), 6.72 ($30^{\circ }$), 6.2 ($45^{\circ }$)	$\pm 20^{\circ }$	−2 to −6	Adjusted	MP CRLH Phase Shifter	$30^{\circ }$, $45^{\circ }$	Minimum variation	LP	–
[62]	2.5	–	0.03	–	–	–	–	Flex Sub AM	–	12% wavelength deviation	LP	–
[64]	8–12	–	–	TX 15.32–21.92, RX 14.87–20.57	–	2.5 to −5.6	–	Flex Sub PCB	PCB 4 cm, TX/RX 3.5 cm	Stable	LP	–
[65]	28	8 × 4	1.0	19.5 ($0^{\circ }$)	$\pm 50^{\circ }$	–	–	Origami, Flex Sub	–	–	LP	–
[66]	10	16 × 16	–	–	–	–	–	Flex PCB	–	–	LP	Space Compatible
[67]	28	16 × 16	1.0	≈3 (element), 58.16 EIRP	–	–	–	Flex Sub	–	–	LP	–
[68]	2.44	Single Antenna	0.33	1.6	–	–	Omni directional	Flex Sub	2 cm to 7 cm	Shift 2.8%	LP	Insensitive (20–$100^{\circ }$C)
[69]	2.5	8 elements	–	–	–	–	–	Flex Sub	– 20.6 cm to 20.6 cm	error ≈0.04λ	LP	–
[70]	10	8 elements	0.48	–	$\pm 60^{\circ }$	–	–	Flex Sub	±12 cm	Stable	LP	–
[71]	5.8	4 × 1	5 MHz	11.9 (flat), 8.7 (bent)	–	–	–	Flex Sub	5 cm	–	–	–
[72]	10.7–12.75	8 × 4	2.05	30.7–34.8	$\pm 60^{\circ }$	≈8.6	–	Flex Sub	5 cm	Stable	Dual Pol.	–
[73]	2.4	1 × 4 subarrays	0.074	≈15	$360^{\circ }$ azimuth	≈7.2	≈17–$23^{\circ }$ (xz–yz)	Balloon Shaped	50 cm	loss ≈ 0.2	LP	–
[74]	2.45	–	240 MHz to 270 MHz	4.8 , 4.3	$80^{\circ }$	–	–	Flex Sub	–	–	LHCP	–

## Data Availability

Not applicable.

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
