# Peer review of "Flexible Phased Antenna Arrays: A Review"

_sensors, 2025, doi:10.3390/s25154690_

Round 1

Reviewer 1 Report

Comments and Suggestions for Authors

This review is comprehensive in content, well-structured, and thorough in coverage. It systematically outlines the technological evolution, key manufacturing techniques, core materials, major challenges, and performance comparisons of Flexible Phased Arrays, while also providing future perspectives. It holds significant reference value. Regarding the manuscript, the following revisions are suggested:

1.Introduction Section Structure:

The introduction is rich in content, covering extensive technical background, historical developments, and key breakthroughs. However, its presentation as a dense, unbroken text block hinders readability and obscures the logical flow. It is recommended to divide this section into clearly defined paragraphs.

2.Undefined Acronym in Figure 1:

The acronym "UVO" used in Figure 1 is not defined in the caption or text. Please provide its full term (Ultraviolet Ozone treatment) upon first use or in the figure caption/list of abbreviations.

3.Lack of Engineering Significance for Figure Data:

The text does not sufficiently interpret the engineering significance of the data presented in figures. For instance, the description of Figure 1(b) – comparing inkjet-printed Silver Nanoparticle ink on Polypropylene substrate with and without UVO surface treatment – is inadequate. It is suggested that the explanation of what this comparison demonstrates be added in the preceding paragraph before introducing the figure.

4.Formula Symbol Definition Issues:

(1)Following Formula (1), it is stated that "Amn represents the excitation amplitude of the mn-th radiating element." However, Amn does not appear in Formula (1). Please verify the correctness of the formula input.

(2)Some symbols in Formula (2) are defined after Formula (1), but definitions for other symbols in Formula (2) are missing. It is recommended to clearly list definitions for all symbols in Formulas (1) and (2) using text or footnotes.

5.Typographical Error:

Line 267: "Tha array factor of an antenna array..." – "Tha" appears to be a typographical error. It should likely be "The".

6.Section 4 Organization and Depth:

(1)While Section 4 covers the major challenges comprehensively, the discussion under subsection 4.1 (Mechanical Deformation) is somewhat scattered. Adding a table summarizing the core principles, advantages, disadvantages, and key metrics of different compensation/design strategies would significantly enhance clarity.

(2)The descriptions in subsections 4.3 (Fabrication Challenges), 4.4 (Thermal Management), and 4.5 (Environmental Challenges) are slightly brief. Expanding these sections with more detail or specific examples is recommended.

7.Performance Comparison Table Analysis:

Table 2 is central to the performance comparison but lacks a brief explanatory note and a concluding analysis. While the table is rich in data, readers are left to interpret it largely unaided. It is strongly suggested to include a concise summary highlighting key trends, trade-offs, and general conclusions drawn from the data in Table 2.

8.Conclusion and Future Outlook Refinement:

Section 6 (Conclusion) could present the core findings of the review more succinctly. While the key challenges and future directions are pointed out, the direction concerning "flexibility and unconventional geometries" is somewhat vague. This point could be elaborated with specific examples or more concrete research avenues within this domain.

Reviewer 2 Report

Comments and Suggestions for Authors

The topic of flexible phased arrays is extremely relevant in the context of the development of unmanned aerial vehicle (UAV) and satellite communications technologies, where lightweight, adaptable and high-performance antenna systems are required.

The review provides a detailed analysis of modern technologies for the production of flexible phased array antennas, including additive manufacturing and hybrid rigid-flexible structures. The review systematizes a wide range of research, including production technologies, materials, deformation compensation methods, and comparative performance analysis of various solutions.

The disadvantages of the review include many specialized terms and abbreviations, which makes it difficult for experts to understand the material.

The descriptive part of the review ends with summary table 2, which lists the technical characteristics of various designs and modifications of flexible phased arrays, but unfortunately there is no critical comparative analysis of the main parameters of phased arrays (broadband, resolution, beam control method, aperture synthesis, performance, etc.) for specific applications, for example, UAVs, spacecraft for probing the earth's surface or geostationary SATCOM repeaters. It is necessary to describe ways to improve the basic technical parameters of phased arrays.

To significantly improve the overview, it is necessary to provide a brief description of each technology, devices, materials, or operating principles in the Abbreviation section.

It is also necessary to additionally decipher all abbreviations in the text of the review (AESA, RF-MEMS, RFID), as well as in Table 2.

After adding additional information and relevant comments, the review can be accepted for publication.

Reviewer 3 Report

Comments and Suggestions for Authors

The paper provides a comprehensive review of flexible phased antenna arrays (FPAs), covering manufacturing techniques, materials, challenges, and performance comparisons. The comments for the paper are as follows:

- The paper has a critical issue regarding its structure and organization. To be more detail, the paper is overly lengthy and repetitive, particularly in Sections 2 and 3, where manufacturing techniques and materials are described in excessive detail without clear thematic progression. An example of overlapping include the discussion about hybrid designs under manufacturing techniques and the part related to material.

- The paper is lack of comparative analysis in Section 5, which is the most critical issue. The stated goal is to perform a "detailed comparative analysis" , but Section 5 contains only a large data table (Table 2) without providing any analytical texts.

- The "Major Challenges" section (Section 4) list the problems but does not prioritize them or propose unified solutions. Besides, the discussion of phase compensation (Section 4.1) is superficial. Equations (1) and (2) are introduced without clarifying their practical implications. Furthermore, key terms (e.g., "array factor") are not defined for a broad audience. 

- The conclusion is weak. It touches upon important points but give a feeling of somewhat disconnected from the wealth of data presented earlier. It only focuses heavily on the limitations of IP and briefly on future geometries.

- Figures (e.g., Fig. 1–4) are poorly integrated into the text, and captions lack technical details. Abbreviations (e.g., LCP, IP) are overused without definitions.

While the topic is relevant, the manuscript lacks critical analysis, suffers from structural disorganization, and omits recent advancements. The authors would need to fundamentally restructure the paper, synthesize existing work into actionable insights, and update references to meet the journal’s standards. Given the extent of revisions required, I recommend rejection in its current form.

Reviewer 4 Report

Comments and Suggestions for Authors

This paper presents a review of flexible Phased Antenna Arrays from the point of view of fabrication, materials and technological challenges.

It is highly recommended to introduce each abbreviation before its first appearance in the text. This method is much more comfortable and understandable for a reader. There is an abbreviation list but it is not complete.

Line 5, please consider suppressing the word “performance” after detailed… as it appears twice in the same sentence. Alternatively, you can rephrase or use a synonym.

Line 7, Please rephrase to suppress repeated words, in this case, “study”

Line 18 AESA is not defined in the abbreviation list

Line 24. Capital letter after period

Line 151. Expected reference after the sentence “Such examples were found in literature”

Line 267. Typo “Tha”

Please provide a reference for equation 1

Please provide a reference for equation (2) or a demonstration of its pertinence in this field.

Line 320 Please clarify the term “micron-sized”

Line 358-368, 426-431: Aircraft and UAV have conformal surfaces rather than flexible. This content is out of the scope of this review, please consider suppressing or justifying how the FPA is used in this scenario (Aircraft /UAV)

Line 20 Please explain what you refer to with “80mm DC power wirelessly at a 1m distance”

Lines 433 to 437 contain an affirmation which is not supported by a reference. Please consider rephrasing as an introduction to desirable challenges that an FPA must endure

The review paper might show graphically how the FPA is compared with alternative solutions at least in terms of frequency Vs power (or other parameter). Thus, a reader can easily visualize where the application focus is placed and which are the current limitations and future challenges of this technology.

Round 2

Reviewer 1 Report

Comments and Suggestions for Authors

The author has conscientiously addressed the feedback and made substantive revisions, resulting in a significant improvement in the paper's quality. While reviewing the current version, I note that certain aspects could still be further refined. Below are my specific recommendations:
(1) In Section 4 Major Challenges, the author has expanded the content by listing existing solutions for each challenge. However, it lacks a further summary of viable future research directions.
(2) The conclusion is presented in a single paragraph without clear transitional phrases, making it difficult for readers to quickly identify key points. I suggest revising this section to enhance the logical flow of the concluding remarks.
I am confident that refining these details will enable the manuscript to better serve researchers in this field.

Author Response

The author has conscientiously addressed the feedback and made substantive revisions, resulting in a significant improvement in the paper's quality.

We would like to thank the reviewer for the positive comment.

While reviewing the current version, I note that certain aspects could still be further refined. Below are my specific recommendations:
(1) In Section 4 Major Challenges, the author has expanded the content by listing existing solutions for each challenge. However, it lacks a further summary of viable future research directions.

As requested by the reviewer we added three sentences trying to address future directions of research, for the specific challenges discussed in the related subsections.
Specifically we added the following three sentences in sections 4.1, 4.2, and 4.4, respectively:

- "Future trends of research to face the mechanical deformation challenges lie in the development of advanced algorithms for phase compensation based on Field Programmable Gate Array (FPGA) technology."

- "An interesting field of research regards the use of AME platforms to achieve precise multilayered structures while maintaining full custom design."

- "Different thermally resilient materials and integrated cooling channels integrated with dynamic cooling systems can be viable solutions for efficient thermal management."

(2) The conclusion is presented in a single paragraph without clear transitional phrases, making it difficult for readers to quickly identify key points. I suggest revising this section to enhance the logical flow of the concluding remarks.

I am confident that refining these details will enable the manuscript to better serve researchers in this field.

The reviewer is right. We tried to restructure the whole section addressing the concern of the reviewer about the logical flow of the section. The modified version is attached below for convenience:

"This paper has analyzed several critical aspects related to the design and development of FPAs. An overview was provided of the most commonly used materials and technologies, as well as the major limitations currently encountered in their implementation.

A first key direction explored involves the use of flexible substrates and AM techniques, such as IP, to realize conductive elements. While this approach is widespread, it typically enables only passive array configurations due to the inherent difficulty in implementing multilayer structures (integrating both conductive and dielectric layers). These multilayers are essential for embedding the control and RF electronics required in active array systems.

Currently, the AME platform Dragonfly by Nanodimensions is the only one that supports 3D-printed multilayer architectures of both conductive and dielectric materials. However, they are limited by the narrow range of compatible dielectric materials and the lack of robust evidence supporting their integration with standard soldering processes, which remain fundamental for high-performance RF systems.

From a manufacturing point of view, a promising alternative lies in the use of rigid-flex PCB technologies. These offer a trade-off between structural flexibility and the reliability of conventional fabrication processes. This area is expected to witness significant developments in the near future, especially as the demand for conformal and adaptive antennas increases.

Another key consideration is the impact of mechanical deformation on electromagnetic performance. The non-planar positioning of radiating elements in flexible arrays can degrade radiation characteristics. Addressing this issue will likely require the development of compensation techniques, such as phase correction and real-time electronic reconfiguration, to maintain optimal performance under deformation.

Furthermore, the design of arrays based on unconventional geometries, such as concentric or concentric-hexagonal lattices, opens new opportunities for performance improvement, particularly in terms of SLL mitigation. However, these configurations pose new challenges for the design and implementation of complex feeding networks, which will require dedicated research efforts.

In conclusion, the data presented in this review can serve as a starting point for identifying the target specifications for various application scenarios involving FPAs. They also provide a useful reference for guiding future research activities in this promising technology field."

Reviewer 2 Report

Comments and Suggestions for Authors

The authors have significantly revised the review and improved it qualitatively.

The article can be accepted for publication in its presented form.

Author Response

The authors have significantly revised the review and improved it qualitatively.

The article can be accepted for publication in its presented form.

We would like to thank the reviewer for the positive comments and the useful advices provided during the reviewing phase.

Reviewer 3 Report

Comments and Suggestions for Authors

The authors have been seriously responsive and have thoroughly addressed all major and minor concerns from the initial review. The critical flaw of the initial submission - a lack of comparative analysis - has been comprehensively rectified with the addition of substantive analytical text that interprets and synthesizes the compiled data in Section 5. Furthermore, the authors have significantly improved the paper's structure for better readability, strengthened the "Major Challenges" section with a new summary table (Table 2) on phase compensation strategies, and rewritten the conclusion to provide an effective synthesis of the review's findings. The manuscript has been transformed into a well-organized, insightful, and valuable review article that is now suitable for publication.

Author Response

The authors have been seriously responsive and have thoroughly addressed all major and minor concerns from the initial review. The critical flaw of the initial submission - a lack of comparative analysis - has been comprehensively rectified with the addition of substantive analytical text that interprets and synthesizes the compiled data in Section 5. Furthermore, the authors have significantly improved the paper's structure for better readability, strengthened the "Major Challenges" section with a new summary table (Table 2) on phase compensation strategies, and rewritten the conclusion to provide an effective synthesis of the review's findings. The manuscript has been transformed into a well-organized, insightful, and valuable review article that is now suitable for publication.

We are very grateful for the reviewer's endorsement. We would like to thank for its very valuable advices that helped us to improve the quality of our work.